# Organizing adult attachment in alternative ways: A qualitative assessment of schemas antithetical to the secure base script

Shannon Maingot[1]*, Marije L. Verhage[1], Carlo Schuengel[1], Theodore E. A. Waters[2], Elja E. J. Meijer[1], Gabrielle Myre[3], Eva C. van Meeuwen[1], Glenn I. Roisman[4], Robbie Duschinsky[5], Marissa D. Nivison[6], Or Dagan[7], Victoria Zhu[2], Marinus van IJzendoorn[8,9], Marian J. Bakermans-Kranenburg[9,10], Sheri Madigan[6], Chantal Cyr[3], Kazuko Y. Behrens[11], Maria S. Wong[12], Elizabeth Meins[13], The Attachment Secondary Processing and Analysis Network of the Collaboration on Attachment Transmission Synthesis (ASPAN-CATS)[¶]

1 Department of Clinical Child and Family Studies, Vrije Universiteit Amsterdam, Amsterdam, Netherlands, 2 Department of Psychology, New York University Abu Dhabi, Abu Dhabi, United Arab Emirates, 3 Department of Psychology, Université du Québec à Montréal, Montréal, Québec, Canada, 4 Institute of Child Development, University of Minnesota, Minneapolis, Minnesota, United States of America, 5 Department of Public Health and Primary Care, University of Cambridge, Cambridge, United Kingdom, 6 Department of Psychology, University of Calgary, Calgary, Alberta, Canada, 7 Department of Psychology, Long Island University–Post, Brookville, New York, United States of America, 8 Research Department of Clinical, Educational and Health Psychology, University College London, London, United Kingdom, 9 Facultad de Psicología y Humanidades, Universidad San Sebastián, Sede Valdivia, Chile, 10 William James Center for Research, ISPA – Instituto Universitário de Ciências Psicológicas, Sociais e da Vida, Lisbon, Portugal, 11 Department of Social and Behavioral Sciences, State University of New York Polytechnic Institute, Utica, New York, United States of America, 12 School of Social Sciences, Communication and Humanities, Endicott College, Beverly, Massachusetts, United States of America, 13 Department of Psychology, University of York, York, United Kingdom

¶ Membership of the Attachment Secondary Processing and Analysis Network of the Collaboration on Attachment Transmission Synthesis Consortium, including group authors and non-author collaborators, is provided in the Acknowledgements.
* s.maingot@vu.nl

## Abstract

How adults make meaning of their childhood experiences with caregivers plays a central role in how they anticipate support, manage distress, and interpret relationships. To understand how adults conceptualize these attachment relationships, the Adult Attachment Interview (AAI) probes for childhood memories of general relationships, day-to-day interactions, and instances of loss, threat, and separation. Across these narratives, the AAI captures expectations of whether caregivers are perceived as an available source of support in times of distress, and whether they serve as a secure base from which to seek comfort and explore the world. Developing a coding system for secure base script knowledge, based on the AAI (AAIsbs), Waters, and Facompré also identified alternative, schema-like representations that seemed to conflict with the secure base script. Before exploring empirical questions regarding these alternative schemas, the present study undertook a qualitative systematic examination of their content in a large, risk-diverse corpus of interviews from 14

---

**Data availability statement:** Coding materials and documentation related to the qualitative analyses may be made available for verification upon request. Access to any interview transcript data is restricted to privacy and ethical concerns, considering the sensitive nature of the interview content. The YODA repository includes a README file that outlines access procedures. All data access requests should be directed to the Faculty of Behavioral and Movement Sciences Data Steward team at VU Amsterdam: research.data.fgb@vu.nl.

**Funding:** This work was supported by a grant from Stichting tot Steun Nederland (stichting-totsteunvcvgz.nl/) to M. Oosterman and C. Schuengel, a grant from the Social Sciences and Humanities Research Council Canada (sshrc-crsh.gc.ca/) (No. 430- 2015-00989) to S. Madigan, a Veni grant by the Nederlandse Organisatie voor Wetenschappelijk Onderzoek Council (nwo.nl) (No.451-17-010) to M. L. Verhage, and a grant from Eunice Kennedy Shriver National Institute of Child Health & Human Development of the National Institutes of Health (United States of America) to Glenn I. Roisman (PI) (nih.gov) (R01HD102035). The content is solely the responsibility of the authors and does not necessarily represent the official views of the National Institutes of Health.

**Competing interests:** The authors have declared that no competing interests exist.

studies. Conducting a thematic analysis, 1,592 AAIs were examined by seven trained coders. This resulted in the identification of three novel themes (Favoritism, Incompetent, and Restrictive), the amendment of three existing schemas (Caregiver Source of Distress, Harsh and Threatening Parenting, and Self-Involved), and the replication of evidence for existing schemas. Together, the diversity and thematic coherence of alternative schemas underscore how relational expectations may be organized around vigilance, distress, or self-reliance, rather than comfort and support. This study contributes to theory-building on relational representations by expanding our understanding of how caregiving relationships are cognitively structured under conditions of relational adversity.

## Introduction

Humans rely on caregivers for protection, regulation, and support in the face of threat. Through repeated interactions with caregivers, children develop expectations about whether proximity to others offers safety or additional risk. Bowlby's [1] attachment theory is rooted in these evolutionary principles, positing that infants seek proximity to an attachment figure when distressed, thereby promoting survival. He framed infant attachment as a motivational system designed to ensure the proximity and availability of caregivers during times of danger. However, when caregiving is inconsistent, proximity may not reliably signal safety and may instead be associated with uncertainty. Bowlby developed this theoretical framework based on observations of children who experienced disrupted caregiving, such as early separation or the loss of a parent [2, 3]. Children form attachment relationships based on their caregiving experiences [4], which helps to understand individual differences in socio-emotional development into adulthood [5–7], as well as in romantic partnerships and child-rearing [8–12]. Exploring how optimal and sub-optimal caregiving experiences—including disruptions to the attachment relationship or harsh and insensitive parenting—impact attachment across developmental stages has remained a key interest in the field.

The Adult Attachment Interview [AAI; 8] was developed to understand how adults conceptualize their past attachment experiences, including those related to separation, loss, and abuse, and how such conceptualizations may guide their caregiving behaviour. The AAI is a semi-structured interview comprised of questions regarding early life experiences with primary caregivers. It often elicits detailed and rich narratives of early caregiving experiences and expectations, but can also produce narratives that are more sparse or difficult to interpret. Indeed, Main and her colleagues [8] showed in a normative risk sample that the qualities of parents' narratives about their attachment experiences, particularly the coherence of their discourse defined in terms of Grice's (1975) [13] maxims, were strongly associated with the quality of attachment relationships their children formed with their parents. Based on these findings, Main et al. [8] developed a classification system for attachment representations of the relationship with primary caregivers, categorizing interviews as secure-autonomous, insecure-dismissing or insecure-preoccupied.

Although the secure-autonomous attachment representation category is indicated by a reasonable level of coherence, insecure-dismissing and insecure-preoccupied categories are assigned when there is low coherence in the narrative throughout the interview. This discovery of the importance of narrative coherence sparked interest in understanding narratives concerning distressing or untoward attachment experiences. In addition, the effects of specific instances of loss, separation, and abuse that sometimes emerge in normative risk samples, and more frequently in at-risk populations, on attachment relationships were not yet well understood. To address these phenomena, Main and Hesse [14] introduced the Unresolved/Disorganized category to capture anomalous discourse on loss and abuse, such as lapses in reasoning, in response to questions posed during the AAI. Interviewees who exhibited more generalized breakdowns in discourse organization during the interview were categorized under Cannot Classify [CC; 15; 16] Building on this framework, [17,18]] proposed the Hostile/Helpless scale for language patterns associated with childhood traumatic experiences in a broader context of consistent, repeated abuse or frightening parenting. These instruments focus primarily on capturing trauma experiences such as physical or sexual abuse [17,14]. Taking an alternative approach, T. E. A. Waters and Facompré [19] sought to delve into the content of AAI narratives on attachment-related experiences through the lens of cognitive schemas. Within a broader system for measuring cognitions on support and availability from primary caregivers (i.e., the secure base script; [H. S. 20], the authors introduced the concept of *alternative schemas.* Positing that repeated experiences with *intrusive* or threatening caregivers undergo the same cognitive process of abstraction and generalization as supportive caregiving experiences, Waters and Facompré [19] suggested that individuals might readily construct schemas related to unsupportive care, which would be reflected in their narratives of early caregiving experiences. These consistently intrusive caregiving narratives, including but not limited to experiences of abuse, still need to be further explored to understand their variety and meaning.

## Scripts and Schemas

Understanding how attachment relationships are represented in the mind has been a primary focus of attachment researchers. Waters and Waters [20] proposed that experiences of secure base use and support, a central component of Bowlby's attachment theory, are internalized and organized as a script-like representation (i.e., the *secure base* script). Ideally, an attachment figure is an available resource from infancy onward, providing a secure base from which the child can explore. With repeated experiences of effective support from primary caregivers in moments of distress, a cognitive script representing this secure base relationship is internalized and consolidated. The secure base script summarizes expectations of the primary caregiver as available in response to distressing events, as well as resolving the stressor and providing comfort, resulting in a return to baseline [H. S. 20, T. E. 21].

To evaluate secure base script knowledge in adults, Waters and Facompré [19] developed the Secure Base Script Scale for use in the Adult Attachment Interview (AAI$_{sbs}$). The AAI$_{sbs}$ rating scale, much like the Attachment Script Assessment [ASA; H. 22], is based on reliable evidence from narratives that reflect knowledge of the secure base script and clarity of this evidence. Participants who describe multiple, whole, or fragmented episodic events in which primary caregivers provided secure base support that follows a script-like structure are thought to conceptualize their relationship with their caregivers in terms of the secure base script. Conversely, generalized descriptions, negative expectations of secure base support, and caregiver failure to meet the child's secure base needs indicate a weak conceptualization of these relationships in terms of the secure base script.

During the development of the AAI$_{sbs}$, schemas that represented generalized expectations distinct from, and seemingly antithetical to, the secure base script were identified. Similar to repeated instances of perceived caregiver support in response to distress, consistent experiences with hostile, unpredictable, or intrusive care may lead to broader theme-like schemas, termed alternative schemas. Repeated and broad descriptions of interactions organized around an alternative schema indicate that the conceptualization of these relationships is incompatible with the basic elements of the secure base script. Expanding on earlier frameworks by Main and Hesse [14] and [17,18], these so-called "alternative

schemas" capture recurring themes of diverse child-caregiver interactions, such as "Harsh and Threatening Parenting" or "Self-Involved." These encompass contexts in which the interviewee's dominant characterization of their relationship with attachment figures is contradictory to the secure base script. In such instances, the caregiver is consistently depicted as a source of distress, threat, anxiety, or frustration rather than as a source of security and support.

Nivison and colleagues [23] found that prospectively documented experiences of abuse and/or neglect were predictive of lower $AAI_{sbs}$ scores, even after controlling for maternal sensitivity and demographic covariates such as biological sex, ethnicity, maternal education, and childhood socioeconomic status. Abusive care from primary caregivers, both mothers and fathers, was uniquely correlated with lower $AAI_{sbs}$ scores: abusive care from other caregivers was not significantly associated. These findings indicate how adverse caregiving experiences from primary caregivers may be uniquely impactful on the development of later attachment script formation. However, this study focused on SBS knowledge and not alternative schemas specifically.

Waters and Facompré [19] identified a preliminary list of 10 specific alternative schemas (Table 1) based on the coding of 15 AAI transcripts (16,955 lines of narrative content). For example, the *Self-Involved* alternative schema was based on recurrent memories reflecting a caregiver who always put their own needs before the child's. The *Role Reversal* alternative schema describes a dynamic in which the child takes on the role of providing support to the parent during times of need, rather than the other way around. While initial schemas were identified, narratives from more risk-diverse samples might be expected to reveal additional schemas.

Empirical work examining the antecedents and outcomes of alternative schemas has yet to follow. Measured using the ASA, secure base script knowledge has been linked to the quality of caregiving received in childhood and to later adult depression symptoms and romantic relationship quality (e.g., [24,25]), underscoring how early caregiving experiences may become encoded in representational scripts that carry consequences across the lifespan. Emerging work examining the impact of maternal childhood abuse versus neglect on infant brain development further suggests that different types of

**Table 1. Original Alternative Schemas Outlined in [19].**

| Alternative Schema | Brief Definition |
|---|---|
| Caregiver as Source of Distress | Caregivers are described as causing distress, often through harsh and threatening behaviors. |
| Dismissive/Unresponsive | Caregiver ignores the child's bids for secure base support, leaving the child to resolve their own distress. |
| Enmeshed/Companionship | Caregiver seeks companionship/comfort from the child; roles blur, though some secure base support may exist. |
| Harsh and Threatening Parenting | Narrative centers around fearing caregiver, monitoring moods, or altering behavior to avoid upsetting them. |
| Child Put in The Middle | Child is asked to moderate or manage conflict between caregivers. |
| Role Reversal | Child fully takes on a parental role; little to no secure base support is provided to the child. |
| Self-Involved | Parent consistently puts their own needs first in the caregiving relationship. |
| Striving for Acceptance/Excellence | Child gains caregiver's attention/affection only by excelling (e.g., academics, sports). |
| Subjugated/Subordinate | Caregiver treats child as subordinate, expecting tasks such as housework. |
| Tit-for-tat | Caregiver and child engage in competing roles, resembling sibling rivalry. |

adverse caregiving experiences may have distinct developmental consequences [26]. This raises the question of whether such experiences also give rise to qualitatively different representational schemas in the adult mind, as captured in the AAI. Whether the perceived relational dynamics captured by alternative schemas, rather than broader classifications, carry differential relevance for such outcomes remains an open and important question. Before such empirical questions can be addressed, a necessary first step is to systematically characterize and refine the set of alternative schemas amongst large, risk-diverse samples.

### The current study

In trying to understand how individuals make meaning of childhood experiences with intrusive or distressing caregivers, it is important to consider that the absence of secure base expectations may not simply indicate a lack of representational structure, but may also reflect the presence of alternative ways of organizing expectations about attachment relationships. The purpose of the current study was to explore the content and scope of alternative schemas in the AAI, as a step towards understanding how characterizations of unpredictable and distressing caregiving in childhood may become internalized as cognitive blueprints in adulthood. We therefore conducted a systematic, semi-open assessment of the preliminary set of alternative schemas within a large, risk-diverse set of 14 samples that are part of the Attachment Secondary Processing and Analysis Network (ASPAN). We first evaluated whether the set of alternative schemas identified by Waters and Facompré [19] also occurred in the current dataset in the form originally described. Second, we examined whether novel themes could be identified, justifying the addition of novel alternative schemas to the set. This broader set of alternative schemas might contribute to further nuancing our understanding of attachment representations, particularly in individuals who experienced adverse caregiving environments.

## Method

### Design

We used a qualitative method, selecting an inductive-deductive, semantic, and 'essentialist/realist" approach for thematic analysis [27]. The inductive-deductive approach was chosen as the current work builds on the original set of themes developed by Waters and Facompré [19], while remaining open to the data that might indicate new themes or a need for changes to existing ones in an inductive manner. The semantic focus was on the level of explicit data (i.e., what has been said by participants in their interviews). Finally, we took an "essentialist/realist" approach that focuses on individual psychologies [27], in that we assume a bidirectional relationship between meaning, experience, and language used by participants without implying an epistemological position.

The current study built on the Collaboration on Attachment Transmission Synthesis (CATS; [28]. Participating CATS authors contributed their datasets, providing demographic data, attachment assessments, and behavioural outcomes to be used for individual participant data (IPD) meta-analyses. For detailed information on the study identification and selection process, see Verhage et al. [11].

### Participants

As part of the ASPAN network, AAI transcripts were included from English (62%), Dutch (28%), and French-speaking (8%) samples. Approximately half of the Dutch and French transcripts were translated into English by a professional translation service. Transcripts were collected from Canada (8%), the Netherlands (27%), the United Kingdom (5%), and the United States (60%). All interviewees were either parents or expecting a child, with 93% of the total sample female and, on average, 28 years old. A total of 1,592 AAI transcripts collected within ASPAN-CATS were coded for secure base script knowledge by our team of seven trained coders, drawn from both normative (38%) and high-risk populations (62%). High-risk status was defined across contributing studies by the presence of one or more risk factors, including adolescent parenthood, parent psychopathology, parental substance abuse, parental history of abuse or trauma, and demographic risk

indicators such as poverty. In terms of ethnicity, 63% of participants identified as White, 30% as Black or African American, 3% as Hispanic or Latino, 1% as Asian, and 2% as other or not reported, with 13% reporting immigrant status. Roughly 50% of participants had completed tertiary education.

## Procedure

In August 2022, coders participated in a three-day training session led by T. E. A. Waters, the primary developer of the $AAI_{sbs}$. The team then coded a reliability set ($n = 100$) composed of Dutch, English, and French transcripts from the ASPAN-CATS corpus from August to December 2022. Coding teams were assigned for each language. Intra-class correlations between coders and Waters for average measures ranged from .84 to .91 and for single measures from .73 to .84. After completing this initial reliability set, the team of seven coders coded transcripts from January 2023 to November 2023 and participated in weekly consensus meetings, with the aim of reaching consensus scores for consensus-coded and double-coded transcripts, as well as to review individual transcripts. Approximately 50% of the transcripts were coded by teams ranging from two to seven coders.

In evaluating the presence of alternative schemas, excerpts deemed relevant were marked as quotations or "chunks" if they represented a particular alternative schema or emerging theme. Following the coding guidelines by Waters and Facompré [19], units of analysis consisted of segments that described a theme-like organization for understanding relationships in ways that are conflictual to the secure base script. Thus, if multiple chunks are recurring throughout the transcript in a theme-like manner, pointing to a world-view-like perspective, an alternative schema may be assigned (See Supplemental Material 1 & 2 for mock-coded transcripts). For example, the *Harsh and Threatening Parenting* alternative schema describes the use of extreme corporal punishment or physical abuse by the caregiver. A portion of text that describes these types of behaviours was marked as a relevant chunk, indicating a potential alternative schema. Although there is currently no benchmark or preset coding guideline to determine how many relevant chunks or quotations need to be present to indicate the presence of an alternative schema, SBS-trained coders assessed whether the collection of relevant chunks pointed to a schema-like worldview that served as the basis for individuals' expectations regarding their caregiver relationship over and above secure base script knowledge.

## Analysis

We conducted a qualitative assessment of alternative schemas coded using the $AAI_{sbs,}$ thematically analysing the interviews using the qualitative analysis software Atlas.ti (Version 24, Scientific Software Development GmbH, 2024) [29]. As the motivation for the current study arose during the larger ASPAN-CATS project of coding AAIs for secure base script knowledge, some of the initial steps required to complete the thematic analysis had already been initiated. A preliminary set of 10 codes, defined by Waters and Facompré [19], served as a starting point for the project, but a data-driven perspective was used, allowing for potential amendments or additions to the preliminary list. A combined inductive-deductive and reflexive approach was thus selected for the thematic analysis, guided by the six steps outlined by Braun and Clarke [27]: Familiarization with the data, generating initial codes, searching for themes, reviewing themes, defining and naming themes, and writing up the final report.

Steps 1–3 of thematic analysis [27] were used from the beginning of the AAI coding process. Although the steps of thematic analysis are listed in a sequential manner, [27, 30] note that working through the phases cyclically is to be expected, to fine-tune candidate themes or continue searching for codes. Although these earlier steps, such as generating initial codes and searching for themes, were initiated during the SBS coding, these stages were revisited throughout the analysis.

With interviews uploaded to Atlas.ti, we identified quotations with selected fragments of the data, which were then coded as a particular alternative schema. We wrote memos and comments with the creation or application of codes and organized these into groups. The first author conducted most of the analysis, while consulting with other members of the

team for guidance or opinions during the review of the themes. This phase of the analysis mainly focused on refinement and potential collapsing of themes with the goal of each theme characterizing a coherent but distinct pattern. Keeping an audit trail of memos and minutes for all consultation meetings, the primary researcher focused on developing definitions with concrete examples for all themes. Next, the first author created succinct and clear titles for novel themes and reviewed the labels of the preliminary schema. These labels were refined until deemed suitable for the final analysis, which provided an extensive description of each alternative schema illustrated by multiple interview chunks. Overall, the first author analyzed 1,108 interview chunks for this aim.

During weekly consensus meetings, coders discussed and reviewed the original set of schemas as well as interview transcripts with evidence of previously unidentified themes. On May 23, 2023, the coding team concluded, after a period of approximately 8 months, that saturation had been reached based on 819 transcripts. Coding continued for the remaining transcripts ($n = 747$) using the updated schema set. Consistent with a reflexive thematic analysis approach [27], inter-rater reliability statistics were not calculated. Reflexive TA does not treat coding as a process of achieving consensus or accuracy across independent coders, but as an interpretive practice through which themes are actively constructed. In the context of qualitative research, Lincoln and Guba [31] suggest four criteria for establishing a study's trustworthiness: credibility, transferability, dependability and confirmability. These criteria were fulfilled in this study by coder training, thick and rich descriptions of data, non-quantified consensual coding, maintaining an audit trail, examining intercoder agreement, and establishing saturation.

## Ethics & privacy

Individual participating studies that have shared their data within the ASPAN project have received IRB approval. Approval has not been sought for the secondary analyses, as it was not required. This study involved a retrospective analysis of previously collected Adult Attachment Interview (AAI) data. The data were accessed for research purposes between 01/08/2022 and 01/10/2025. Authors who shared data for the multi-study data consortium were asked to provide manuscripts without personal information, but occasionally, some personal information was still present. Because the interviews are open-ended, participants occasionally mentioned names of people or places during the interview. Transcripts were given different ID numbers than other data received to minimize linking opportunity, and the key was held only by one of the primary investigators. Junior researchers were then involved in the de-identification of interview materials and, therefore, had limited, incidental exposure to identifying information during data preparation. All identifying information was removed prior to coding. The analytic dataset contained only coded participant identifiers, and the authors did not have access to identifiable information during or after data analysis.

## Results

Content note: These quotes reflect the lived experiences of individuals affected by harsh caregiving and/or trauma and may include descriptions of distress or painful events. While these voices are essential to understanding alternative schemas, we acknowledge that this content may be difficult to read.

The thematic analysis yielded three clusters: (1) pre-existing themes earlier identified by Waters and Facompré [19] that also occurred in the ASPAN-CATS dataset, (2) pre-existing themes that were ultimately amended, and (3) novel themes. Definitions outlined by Waters and Facompré of seven pre-existing themes were found to be represented in interview chunks within multiple transcripts, with little to no evidence of varying conceptualizations to the original schemas. These recurring themes/schemas, with additional interview chunks drawn from ASPAN-CATS, are listed in Table 2. However, the thematic analysis led to amendments of three pre-existing schemas (Caregiver Source of Distress, Harsh and Threatening Parenting, Self-Involved/Neglect) as well as three novel themes (Favoritism, Restrictive, and Incompetent). Following the saturation meeting, no additional novel themes were identified. The novel and amended themes are described below, and supporting interview chunks are presented.

**Table 2. Alternative schemas recurring in ASPAN-CATS Sample.**

| Alternative Schema | Interview Excerpt |
|---|---|
| Dismissive/Unresponsive | "A lot of the times. He didn't want to talk to us. Um, what else. As a kid, trying to reach to him, he was like pushing me away." |
| Child Put in The Middle | "I was scared to go tell my father… I had this fear that she [MOTHER] would hurt my father. And so, I never told him." |
| Enmeshed/Companionship | "I was little, I took care of him like a friend would… I was about seven or eight giving him advice… He'll talk to us, tell us." |
| Parent Demands Excellence/ Striving for Acceptance | "She wasn't a teacher… if I didn't pick them up like on the first go through, she would get really angry…" |
| Subjugated/Subordinate | "I sort of became a substitute for my step-dad… she sort of treated me like, you know, can you take care of your little sister…" |
| Role Reversal | "I think that both my parents have always spoken very openly about their own problems with me… I have been more supportive to them." |
| Tit-for-Tat | "Um, it was always confusion… when I got a little older, I was going off on him, he was going off on me." |

## Amendments

**Caregiver source of distress.** The Caregiver Source of Distress schema was originally intended for caregivers described as being causally linked to the child's distress and for narratives of recurrent hostile and threatening caregiving behaviours [19]. However, this broad definition posed three issues. First, most alternative schemas are defined as a source of distress or anxiety for the child due to volatile, intrusive and/or unpredictable caregiving. Based on this understanding of schemas, the definition given to Caregiver Source of Distress could (to some extent) be applied to all alternative schemas. Second, the current definition showed some overlap with another pre-existing schema, Harsh and Threatening Parenting. The main characteristics of a Harsh and Threatening Parenting schema are a description of consistent fear of the parents and likely recurrent abuse. The overlap between the schemas hampered assigning the most appropriate schema. Third, further confusion during the coding process stemmed from the original guidelines highlighting Caregiver Source of Distress as a catch-all: "In cases where more than one alternative schema emerges, coders may have difficulty picking a dominant schema and thus assign the more general parent source of distress" ([19], p. 41). As a result, different experiences—ranging from sexual abuse to anxiety in the home or a break of trust with a caregiver—were all collapsed under Caregiver Source of Distress. For example, both of the following chunks were originally coded under Caregiver Source of Distress, despite representing qualitatively different dynamics.

"When my step-father um… was, would sex molest me. I cried after…"

"He had these old pans that he was, I don't know, preparing breakfast. And he would throw the frying pan across the room. He didn't ever hit anybody. He wasn't trying to hit at somebody, but he was, he was so mad he was just throwing things around and it was loud. I didn't understand what was going on and it was scary."

Following guidelines from [27, 30] for thematic analysis, our goal was to ensure minimal overlap between themes. This led to the following amendments to the alternative schemas Caregiver Source of Distress and, as detailed below, Harsh and Threatening Parenting.

We defined Caregiver Source of Distress as dynamics in which the caregiver's behaviour elicits general distress, frustration, or anxiety. For example, a chaotic parent who struggles to manage the home may bring anxiety to the child. The child may describe feelings of annoyance, dislike or distrust toward their parent, possibly due to a lack of predictability. The following chunk describes unease when the caregiver is home, even though the child is not directly threatened by the parent's behaviours. Although the adult, in retrospect, appears to have suffered from the consequences of the parent's frustrating behaviours, these caregiving behaviours did not directly target the child and are not described as fear-provoking.

"That's similar to my feeling that when she wasn't there that everything went much easier. And yes, that when she came home things were just worse. We had a lot of fights in the apartment, everyone arguing with each other. And if she wasn't there then we didn't have that."

**Harsh and threatening parenting**

In line with the amendments described above, any descriptions of a direct "attack" or threat by the caregiver through fear-provoking behaviours fall under Harsh and Threatening Parenting. A lack of predictability is described where the child is the direct target of the intrusive or volatile behaviour from the caregiver. Any form of recurrent abuse (sexual, verbal and/or physical) would fall under this schema.

"I had a good.. uh, relationship with my uncle 'til, when I was staying with him. Oh yeah, my uncle. When I was staying with him, uh...he had had sex with me."

Recurring references to physical "punishment" that are extreme considering culture and time period also point to a Harsh and Threatening Parenting schema.

"I always got a whoopin' for every single thing that I did. Everything that I did, I got a whoopin' for. I got a whoopin' for, you know-- just sayin' the wrong thing or, - -of course, you know, cuttin' up in school, -- uh, not doin' the chores that I was supposed to do."

**Self-involved/Neglect**

This schema, originally labelled Self-Involved, was defined as follows: "the child comes to learn that the needs of the parent always come first" [19, p. 44]. The child has learned not to seek out secure base support in times of need due to the lack of regard from the caregiver. Based on chunks providing additional nuances, we added the following details: the caregiver is unconcerned with the child's daily life experiences, does not play with the child, or puts no effort into their relationship. The following example illustrates how a participant perceived their caregiver as uninterested in spending time together.

"Because I knew he was comin', like, on [Holiday] or birthdays (MmHm), [Holiday 2]. You know? 'Cause the only thing he would do was come pick us up and take us to his sister's house. (MmHm.) Drop us off, or whatever. I mean, that, that was fine, but I wanted to spend time with him. (MmHm.) You know? And, I, I, I thought that he didn't wanna spend time with me so I didn't mention it."

In some cases, parents' perceived self-involvement may reach the level of neglect, when even basic instrumental care (hygiene, food, cleanliness, and safety of home, life threatening situations, etc.) is not provided by the caregiver.

"No, but those are things I heard later from other people. Who were like, well you were actually neglected as a child. Emotionally neglected, well that in [Place], cockroaches walked over us while we were sleeping for example. Well those are things that happened there of course but, look my parents were young too."

Based on such chunks detailing instances of serious neglect or a very high level of self-involvement, we changed the label of the schema to Self-Involved/Neglect.

Finally, we came across many chunks that pointed to two specific subsets of Self-Involved/Neglect, specifying either addiction or a romantic relationship as being the reason that a child's needs are secondary to the caregivers.

*Addiction.* For the Addiction subtheme, the caregiver engages in substance abuse, alcohol abuse, or gambling to the point that the child's basic care is neglected.

"Confused ah... I guess just ah.... him falling into the drug category, doing his drugs or whatever. Ah... I guess that would make anybody confused not knowing whether he should ah... be there for us or do what he got to do to.... to get his drugs, or be there for his drugs, or whatever. I don't know."

"A junkie. Cause that's what I feel about her. She ain't...she...she let the drugs take'er over and that caused him not to be a woman, or a mother, or a wife, or whatever. So..."

*Romantic Relationship.* This subtheme was present when the participant described a parent prioritizing a romantic relationship over the physical and/or emotional needs of the child.

"She put her men before her kids. And when I say, "irresponsible," she would just say, 'make sure he ate, you know, before we did?' And that's, you know, really...really irresponsible and inconsiderate, [...]. I mean it was a hurting feelings. Me and my sibling, you know, we ate but it was like he got to eat first? You know? And it shoulda been the other way around or it was always her men first! It was never us, so."

## Novel themes

**Favoritism.** This theme encompasses descriptions of a caregiver's consistent preferential behaviour towards a particular or towards other siblings. The child perceives that one or more siblings can do no wrong in the parent's eyes. Perhaps one child is seen as the "golden child" in the family, leaving the participant with a feeling that they in turn lacked attention and care.

"I just felt like, my sister was the golden child, whatever she did was, (in high, sarcastic voice:) thrilling, whatever,you get some real angst going on, I tell you, on this subject, whatever she wanted she seemed to get, at the expense of maybe things that I wanted because she demanded it and she got it."

In this interview chunk, the participant describes having felt a marked difference between the treatment of the younger "spoiled" sibling and the punishment or discipline they received. However, examples of Favoritism need not only involve cases of discipline, but may simply detail preferential treatment leaving a sense of lack of affection, attention, or availability for the participant as a child.

"He used to always make me mad because he always took my little sister around with her whenever he was going to the grocery store, or going shopping for clothes, or whatever. And what made me so mad is because he used to sneak and do it. (Okay.) So maybe he'll wake her up early and they'll just leave. By the time I'm up, they'll be gone. And then, she used to always come back with like little things and something that he bought. He'll buy her things and whatever

because she went. That's why I always used to like to go with him, but....for some reason... He...he used to say, it's because he didn't want to comb my hair. It took too much time or whatever, but I used to...sometimes, I used to cry, and sit in the window, and wait for them to come back."

*Scapegoat*. We specified a "Scapegoat" subcategory under the Favoritism alternative schema, for interviews that recurrently describe events where they felt as though undue blame was placed on their shoulders. The subtheme describes the child-caregiver relationship as being framed in the caregiver's contrasting treatment between siblings, but differs in that overwhelmingly negative attention was placed on the participant. The child may self-identify as the "problem child". For example,

"Oh he was always screaming and shouting at me... everything's…my fault."

"She was always... […]...shouting at me......anything happened in the house it was my fault (right)... if the refrigerator broke it was me... not one of the boys."

## Restrictive

This theme regards caregivers who restricted or discouraged the child from exploration, play, activities, and/or self-expression. In doing so, they hindered rather than supported exploration, contradicting the secure base script. The participant may express a certain frustration, feeling stifled, a lack of freedom to be themselves or to engage in age-appropriate activities (e.g., play outside, sleepovers, after-school programs) or pursue their own interests. Overall, the child did not feel supported or may have expressed feeling actively discouraged in their personal development.

"We used to fight quite a bit all through high school and everything about what I would be allowed to do or not allowed to do and I'd always refer to you know, well the other kids are doing this and my step-mom was like ahh, you know, well you're going to get sick, or you're gonna you know, go break a leg or whatever so."

"Because my sister, growing up, she was [Number] years older than me and she was really promiscuous in high school, so growing up, I guess he [Father] thought that I was gonna be the same way, which I wasn't. And all I wanted to do really was have a social life, I didn't want to go out and have sex and all that and, I feel like he took that away from me, he wouldn't let me go outside, it felt like I was in jail. And then you know, when you young, that age, you trying to get your identity and I couldn't really find my identity. I used to hate going home because every room, it would be a child, I didn't have any privacy. And I hated him for that!"

## Incompetent

With this schema, the participant calls into question the competence of the attachment figure as a caregiver, to the point that the caregiver's parenting abilities are consistently called into question or seen as outright lacking.

"But he ah....he...I don't think he knew nothing about raising no...no children."

An Incompetent alternative schema may be present where there are multiple AAI$_{sbs}$ codes assigned for a negative Stronger and Wiser secure base expectation, indicating that the caregiver's authority and competence are brought into question. A negative Stronger and Wiser secure base expectation is coded when the interviewee describes considerable doubt in the caregiver's ability to resolve a conflict [19]. The participant may see the caregiver as incapable of parenting, despite their efforts, and as undeserving of their respect.

"When I began to ask questions, and poke and prod at him, he began to turn mean, so it's like I don't know why you're being mean but I'm just asking questions so, then that kind of made my relationship with him, I didn't see him as, you know, Dad anymore, it's like this man who, umm, what's the word… whom I didn't have much respect for, I'll put it that way, I didn't have a whole lot of respect for him after that cause everything."

The child may have looked down on their caregiver. A participant with an Incompetent schema may make derogatory comments regarding the caregiver's parenting skills.

"[…] I've forgiven her many times for the mistakes she's made, but I'm disappointed every time. Which led to us arguing again, which led to me forgiving her again, and then doing the same thing again. And that's why it's very changeable, because I don't know how I should feel towards her. Whether I regard her as my own mother or not."

## Discussion

The present qualitative study assessed to what extent the alternative schemas in the AAI$_{sbs}$ coding system as outlined by Waters and Facompré [19] could be identified in a large and diverse corpus of AAIs. It also explored whether amendments to these schemas were required and whether new schemas could be identified. Following the thematic analysis, we proposed three amendments to the originally defined schemas (Caregiver Source of Distress, Harsh and Threatening Parenting, and Self-Involved/Neglect), and three novel schemas (Incompetent, Restrictive, and Favoritism). Seven of the ten original schemas reoccurred without amendments in the ASPAN-CATS sample. Furthermore, the thematic analysis contributed to greater detail to the coding guidelines of existing schemas and addressed overlap among different schemas.

### Reaching saturation

We reached a saturation of themes for alternative schemas during coding of this large corpus of AAIs, further increasing confidence in the schemas identified. After a period with no additional themes emerging beyond Favoritism, Restrictive, and Incompetent, the coding team conducted a saturation check at 814 transcripts coded. No evidence for additional novel themes emerged beyond this point. Although this may indicate that the sample was sufficiently diverse to cover the variation present in Western, English-dominant samples, novel themes may occur in other linguistic contexts or cultural populations. The attachment field has been widely criticized for its assumptions of universality in attachment [i.e., 32; 33]. Indeed, it is possible that certain elements assumed to be central to secure base provision in Western society may not necessarily be present in other cultures.

Beyond sample diversity, the cultural context in which caregiving takes place may shape the specific content of alternative schemas. What constitutes fear, distress, or a failure of secure base provision is partly culturally situated. During coding, for instance, corporal punishment was not automatically coded as Harsh and Threatening Parenting unless the narratives reflected fear, an approach that attempted to account for cultural variation in disciplinary norms. Future work examining schema content across diverse cultural settings would be a valuable direction.

### The value of differentiation

The current qualitative analysis shows that themes that are indicative of alternative schemas initially identified by Waters and Facompré [19] are also present in AAIs from the set of 14 risk-diverse samples collected within ASPAN-CATS. The alternative schema construct could be meaningfully parsed into distinct subtypes, as evidenced by the amendments, the emergence of three novel schemas and the recurrence of existing schemes.

Although this differentiation adds nuance to the conceptualization of alternative schemas, the scientific value of this finer-grained parsing now needs to be determined. One question moving forward is whether identifying specific types of

schemas—for example, Role Reversal and Self-Involved—enhances the precision of predicting or explaining relevant developmental, health, or psychological outcomes. It may be that these distinct schemas are differentially associated with particular individual or contextual factors, revealing patterns that would remain obscured when examining only the overall presence of alternative schemas. For instance, some schemas may be more prevalent in particular populations, which could inform future directions for research and potentially clinical practice. At the same time, the differentiation may contribute more to the reliability of the overall alternative schema score (1 on the 9-point $AAI_{sbs}$ scale) than to conceptual clarity—akin to the approach used in coding disorganized attachment [34]. Subtypes of disorganized behaviour can inform a reliable scale score but are not analyzed at this level of granularity, given unclear reliability and low base rates of the specific indicators of disorganized attachment. Similarly, attaining reliability on the distinct types of alternative schemas rather than the overall score may prove challenging.

The current study was conducted in the context of discovery, as the identification of themes remained an open process throughout coding. Future trainings with coders examining new samples may offer an opportunity to establish reliability on the alternative schema subtypes. Ultimately, further research is needed to determine whether these specific schema types can be reliably coded and have unique correlates, or whether they primarily serve to improve the reliable identification of alternative schemas as a whole. In either case, the broader aim remains to better capture and understand adults' representations of disruptive attachment experiences with primary caregivers.

## Theoretical meaning of alternative schemas

Beyond reliability and coding utility, capturing distinct schemas may carry theoretical meaning in its own right. These schemas reflect a qualitatively distinct representational organization of attachment-relevant experiences with primary caregivers. Rather than simply indicating the absence of secure base script knowledge, they outline in detail the recurring expectations of how caregiving relationships function under conditions of fear, distress, role confusion, unpredictability, or emotional unavailability.

This distinction may offer a complementary approach to understanding non-autonomous attachment classifications. With the move to the level of representation [8], security and insecurity in adult representations of attachment are assessed on the basis of discourse coherence and attentional strategies during the AAI, where dismissing and preoccupied classifications describe ways in which these discourses of attachment-related experiences are limited and distorted. Alternative schemas offer a different lens: rather than how narratives are organized, they capture the explicit content of attachment-relevant experiences as recalled during the AAI. Mapping the alternative schemas onto the non-autonomous classifications may help capture which expectations and behaviors underlie preoccupied or dismissing attachment strategies. For instance, dismissing representations may not only manifest through implicit discourse indicators such as idealization or lack of memory [35], but also through explicit, temporal-causal, relational content described by schemas such as Self-Involved/Neglect or Incompetent. Similarly, discourse indicators for preoccupied (e.g., passive speech, difficulty moving on from topics) may be further specified through schemas such as Enmeshed or Role Reversal. Examining this additional layer of specificity, detailing the particular parenting behaviors that led to the caregiver being perceived as a source of distress rather than support, may hold relevance for both research and clinical practice. Empirical work directly examining the overlap between alternative schemas and AAI classifications is needed to test these propositions.

## Antecedents, consequences, and utility of the alternative schema

With this qualitative study, we have further developed a tool to classify how adults may describe negative attachment experiences from childhood. Future studies may begin to quantitatively assess how alternative schemas develop and what their consequences are. If reliable coding turns out to be feasible, studying the developmental origins of alternative schemas and secure base scripts may broaden our understanding of cognitions around disruptive attachment experiences beyond the constructs that are currently in use, in particular unresolved trauma and hostile-helpless attachment

representations. Incremental validity of the alternative schemas over traditional AAI ratings and classifications may be a crucial next step.

The relevance of such questions is underscored by existing work on secure base script knowledge more broadly. Dagan and colleagues (2021, 2024, 2025) [24,36,37] showed that greater secure base script knowledge was associated with better emotional and physical health, as well as higher romantic relationship quality, specifically lower perceived hostility from a romantic partner. Furthermore, secure base script knowledge mediated the relation between maternal and paternal sensitivity in childhood and romantic partner hostility and emotional health. This finding on secure base script knowledge aligns with previous research, supporting the idea that this form of attachment representation, built on repeated experiences of sensitive caregiving [23,25,38,39], may influence the processing of interpersonal information and behaviours later in life [36,40–43]. Considering that alternative schemas similarly reflect expectations based on childhood experiences with primary caregivers, it remains an open question how these schemas, which specify content that conflicts with elements of secure base provision, may also impact adult functioning.

The alternative schemas may offer both research and practical utility by providing detailed information on how adults make sense of caregiving relationships and organize their broader expectations of others. What it means, representationally, to organize expectations around experiences of threat and fear may be vastly different from experiences of taking on the caregiving role: one might expect that an individual describing the Harsh and Threatening Parenting schema perceives the world as a particularly hostile place, in contrast to someone with a schema scored as Enmeshed. Gaining a better understanding of the antecedents and consequences of such cognitive schemas may therefore be a promising avenue for both researchers and clinicians. In this regard, Raby and colleagues [44] showed that secure base script knowledge improved following intervention in parents referred by Child Protective Services, and that this increase mediated improvements in parental sensitivity seven years later. These findings support the view that cognitive scripts are malleable and can be modified through new experiences ([1, 45, 21, but see 46], for null effects on script change following intervention, which the authors attribute to possible sleeper effects). The current study offers further description and delineation of alternative schemas in a risk-diverse sample, providing a first step towards future research on whether cognitive scripts centered on themes of fear and unpredictability may also show malleability and openness to change.

## Limitations

While the current work provided added detail to the $AAI_{sbs}$ coding schema for identifying alternative schemas, it remains an exploratory study. The interview coding was conducted in the context of discovery, and reliability was not verified for the alternative schema subtypes. While reliability is not a necessary construct for the nature of a qualitative, thematic analysis (but see [47], for a contrasting perspective), we do remain cautious in the implications of our findings. Before testing the many hypotheses generated above, establishing the psychometric properties of the schemas is an important first step.

Additional features of this study may limit the generalizability of the findings. Although the sample was risk-diverse, it remains Western, and theme saturation may not extend to more geographically diverse populations. The thematic analysis thus does not explicitly account for whether different caregiving norms may give rise to additional or differently nuanced schemas. Furthermore, although the sample included both mothers and fathers, and schema types did not appear to differ by gender on initial inspection, the sample was predominantly female. Findings may therefore not fully capture the range of schemas that could emerge in male caregivers. Finally, although transcripts were collected and coded across three languages, the preliminary schema definitions and coding guidelines were developed in English. This may limit how well subtle cultural or linguistic nuances are captured in childhood narratives. The AAI itself was originally developed in English, and translations may carry subtle differences in how questions are phrased or interpreted, potentially influencing the narratives elicited and, in turn, the schemas identified.

## Conclusions

Alternative schemas appear to capture details present in the narratives of adults who, in the context of the Adult Attachment Interview, report about childhoods with little secure base support. The $AAI_{sbs}$ coding scheme for alternative schemes was successfully applied to a large, diverse set of AAIs. Through application of the steps of thematic analysis, three schemas were amended, providing greater nuance to their definitions, and three novel schemas were identified based on recurring theme-like narratives. Although the analysis added precision to the $AAI_{sbs}$ coding system, there is still much to learn about these cognitive structures and their precursors and consequences. The coding of alternative schemas may provide a promising approach to understanding with greater nuance individuals who perceive a lack of secure base in childhood. Alternative schemas may help capturing complexities of how adults recall dynamics with caregivers who were meant to be a haven of safety, but failed.

## Supporting information

**S1 File. Mock adult attachment interview- Role reversal alternative schema.**
(PDF)

**S2 File. Mock adult attachment interview- Harsh and threatening parenting alternative schema.**
(PDF)

**S3 File. Alternative schema coding guidelines with novel and amended themes.**
(PDF)

## Acknowledgments

This work was conducted within the Attachment Secondary Processing and Analysis Network of the Collaboration on Attachment Transmission Synthesis (ASPAN-CATS) research consortium. We gratefully acknowledge the contributions of consortium the co-authors, the group authors who reviewed and approved the final manuscript, as well as the non-author collaborators who contributed to the broader ASPAN-CATS collaboration. The co-leads of ASPAN-CATS are Marije L.Verhage (m.l.verhage@vu.nl) and Carlo Schuengel (c.schuengel) at Vrije Universiteit Amsterdam. Co-Authors: Shannon Maingot (Vrije Universiteit Amsterdam), Marije L. Verhage (Vrije Universiteit Amsterdam), Carlo Schuengel (Vrije Universiteit Amsterdam), Elja E. J. Meijer (Vrije Universiteit Amsterdam), Eva C. van Meeuwen (Vrije Universiteit Amsterdam), Theodore E. A. Waters (New York University Abu Dhabi), Victoria Zhu (New York University Abu Dhabi), Glenn I. Roisman (Institute of Child Development, University of Minnesota), Gabrielle Myre (University of Quebec in Montreal), Chantal Cyr (University of Quebec in Montreal), Robbie Duschinsky (University of Cambridge), Marian J. Bakermans-Kranenburg (University Institute of Psychological, Social and Life Sciences Lisbon; Facultad de Psicología y Humanidades, Universidad San Sebastián, Sede Valdivia, Chile), Marinus van IJzendoorn (University College London), Marissa D. Nivison (University of Calgary), Sheri Madigan (University of Calgary), Or Dagan (Long Island University-Post Campus), Kazuko Y. Behrens (State University of New York Polytechnic Institute), Maria S. Wong (Endicott College), Elizabeth Meins (University of York). Group Authors: Jude Cassidy (University of Maryland), Brent Finger (Montana State University Billings), Lee Raby (Department of Psychology, University of Utah). Non-Author Collaborators: Mary Dozier (University of Delaware), Esther Leerkes (University of North Carolina at Greensboro), George Tarabulsy (Université Laval Québec), Doug Teti (Pennsylvania State University; Edna Bennett Pierce Prevention Research Center)

## Author contributions

**Conceptualization:** Shannon Maingot, Marije L. Verhage, Carlo Schuengel, Theodore E.A. Waters, Elja E. J. Meijer, Gabrielle Myre, Eva C. van Meeuwen, Glenn I. Roisman, Robbie Duschinsky.

**Data curation:** Shannon Maingot, Marije L. Verhage, Carlo Schuengel, Elja E. J. Meijer.

**Formal analysis:** Shannon Maingot, Marije L. Verhage, Carlo Schuengel, Theodore E.A. Waters, Elja E. J. Meijer, Gabrielle Myre, Eva C. van Meeuwen.

**Funding acquisition:** Marije L. Verhage, Carlo Schuengel, Theodore E.A. Waters, Glenn I. Roisman, Chantal Cyr.

**Investigation:** Shannon Maingot, Marije L. Verhage, Carlo Schuengel, Theodore E.A. Waters, Elja E. J. Meijer, Gabrielle Myre, Eva C. van Meeuwen, Marissa D. Nivison, Or Dagan, Victoria Zhu.

**Methodology:** Shannon Maingot, Marije L. Verhage, Carlo Schuengel, Theodore E.A. Waters, Elja E. J. Meijer, Gabrielle Myre, Eva C. van Meeuwen.

**Project administration:** Shannon Maingot, Marije L. Verhage, Carlo Schuengel, Elja E. J. Meijer.

**Resources:** Marije L. Verhage, Carlo Schuengel, Glenn I. Roisman, Marinus van IJzendoorn, Marian J. Bakermans-Kranenburg, Chantal Cyr, Kazuko Y. Behrens, Maria S. Wong, Elizabeth Meins.

**Supervision:** Marije L. Verhage, Carlo Schuengel, Theodore E.A. Waters.

**Visualization:** Shannon Maingot, Marije L. Verhage, Carlo Schuengel, Theodore E.A. Waters.

**Writing – original draft:** Shannon Maingot.

**Writing – review & editing:** Marije L. Verhage, Carlo Schuengel, Theodore E.A. Waters, Elja E. J. Meijer, Gabrielle Myre, Eva C. van Meeuwen, Glenn I. Roisman, Robbie Duschinsky, Marissa D. Nivison, Or Dagan, Victoria Zhu, Marinus van IJzendoorn, Marian J. Bakermans-Kranenburg, Sheri Madigan, Chantal Cyr, Kazuko Y. Behrens, Maria S. Wong, Elizabeth Meins.

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
