## [Decision Letter · Decision Letter 0]

24 Mar 2026

PONE-D-26-03584Organizing Adult Attachment in Alternative Ways:

A Qualitative Assessment of Schemas Antithetical to the Secure Base ScriptPLOS One

Dear Dr. Maingot,

Thank you for submitting your manuscript to PLOS ONE. After careful consideration, we feel that it has merit but does not fully meet PLOS ONE’s publication criteria as it currently stands. Therefore, we invite you to submit a revised version of the manuscript that addresses the points raised during the review process.

We look forward to receiving your revised manuscript.

Kind regards,

Tobias Otterbring

Academic Editor

PLOS One

Journal Requirements:

“This work was supported by a grant from Stichting tot Steun Nederland (stichtingtotsteunvcvgz.nl/) to M. Oosterman and C. Schuengel, a grant from the Social Sciences and Humanities Research Council Canada (sshrc-crsh.gc.ca/) (No. 430-2015-00989) to S. Madigan, a Veni grant by the Dutch Research Council (nwo.nl) (No. 451-17-010) to M. L. Verhage, and a grant from the National Institutes of Health to Glenn Roisman (nih.gov)(R01HD102035). Research reported in this publication was supported by the Eunice Kennedy Shriver National Institute Of Child Health & Human Development of the National Institutes of Health under Award Number R01HD102035 to GIR. The content is solely the responsibility of the authors and does not necessarily represent the official views of the National Institutes of Health.”

4. In the online submission form you indicate that your data is not available for proprietary reasons and have provided a contact point for accessing this data. Please note that your current contact point is a co-author on this manuscript. According to our Data Policy, the contact point must not be an author on the manuscript and must be an institutional contact, ideally not an individual. Please revise your data statement to a non-author institutional point of contact, such as a data access or ethics committee, and send this to us via return email. Please also include contact information for the third party organization, and please include the full citation of where the data can be found.

5. One of the noted authors is a group or consortium “The Attachment Secondary Processing and Analysis Network of the Collaboration on Attachment Transmission Synthesis (ASPAN-CATS)”. In addition to naming the author group, please list the individual authors and affiliations within this group in the acknowledgments section of your manuscript. Please also indicate clearly a lead author for this group along with a contact email address.

7. We note that this data set consists of interview transcripts. Can you please confirm that all participants gave consent for interview transcript to be published?

If they DID provide consent for these transcripts to be published, please also confirm that the transcripts do not contain any potentially identifying information (or let us know if the participants consented to having their personal details published and made publicly available). We consider the following details to be identifying information:

- Names, nicknames, and initials

- Age more specific than round numbers

- GPS coordinates, physical addresses, IP addresses, email addresses

- Information in small sample sizes (e.g. 40 students from X class in X year at X university)

- Specific dates (e.g. visit dates, interview dates)

- ID numbers

Or, if the participants DID NOT provide consent for these transcripts to be published:

- Provide a de-identified version of the data or excerpts of interview responses

- Provide information regarding how these transcripts can be accessed by researchers who meet the criteria for access to confidential data, including:

a) the grounds for restriction

b) the name of the ethics committee, Institutional Review Board, or third-party organization that is imposing sharing restrictions on the data

c) a non-author, institutional point of contact that is able to field data access queries, in the interest of maintaining long-term data accessibility.

d) Any relevant data set names, URLs, DOIs, etc. that an independent researcher would need in order to request your minimal data set.

For further information on sharing data that contains sensitive participant information, please see: https://journals.plos.org/plosone/s/data-availability#loc-human-research-participant-data-and-other-sensitive-data

If there are ethical, legal, or third-party restrictions upon your dataset, you must provide all of the following details (https://journals.plos.org/plosone/s/data-availability#loc-acceptable-data-access-restrictions):

1. A complete description of the dataset

2. The nature of the restrictions upon the data (ethical, legal, or owned by a third party) and the reasoning behind them

3. The full name of the body imposing the restrictions upon your dataset (ethics committee, institution, data access committee, etc)

4. If the data are owned by a third party, confirmation of whether the authors received any special privileges in accessing the data that other researchers would not have

5. Direct, non-author contact information (preferably email) for the body imposing the restrictions upon the data, to which data access requests can be sent

8. We note that there is identifying data in the Supporting Information file “SUPPLEMENTAL FULL_Altschemas_formatting.pdf”. Due to the inclusion of these potentially identifying data, we have removed this file from your file inventory.

Prior to sharing human research participant data, authors should consult with an ethics committee to ensure data are shared in accordance with participant consent and all applicable local laws.

-Location data

Please remove or anonymize all personal information “Age”, ensure that the data shared are in accordance with participant consent, and re-upload a fully anonymized data set. Please note that spreadsheet columns with personal information must be removed and not hidden as all hidden columns will appear in the published file.

Additional Editor Comments:

Dear authors,

Your paper has now been reviewed by two experts in this topic domain. Both the reviewers generally like what you have done, although their assessments diverge slightly, with one of them recommending a major revision and the other one suggesting a minor revision. Please see their detailed comments below. Based on the many thoughtful remarks provided by the reviewers, I am willing to move your manuscript into a second round of reviews. This will be a major revision with a clear path toward publication as long as you carefully address the concerns raised by the reviewers, with corresponding edits in the manuscript.

Please resubmit your revision at your earliest convenience and no later than 60 days from now. If you need additional time, please let me know at tobias.otterbring@uia.no.

Kind regards,

Tobias Otterbring

Handling Editor, PLOS ONE

Reviewers' comments:

Reviewer's Responses to Questions

**Comments to the Author**

1. Is the manuscript technically sound, and do the data support the conclusions?

Reviewer #1: Yes

Reviewer #2: Yes

2. Has the statistical analysis been performed appropriately and rigorously? 

Reviewer #1: N/A

Reviewer #2: N/A

3. Have the authors made all data underlying the findings in their manuscript fully available?

Reviewer #1: No

Reviewer #2: Yes

4. Is the manuscript presented in an intelligible fashion and written in standard English?

Reviewer #1: Yes

Reviewer #2: Yes

5. Review Comments to the Author

Reviewer #1: “Organizing the attachment to alternative ways: A qualitative assessment of schemas antithetical to the secure base script”

The manuscript presents a qualitative reanalysis of data obtained through the Adult Attachment Interview (AAI). I find the paper very interesting and valuable, particularly because it demonstrates the diversity of behavioral patterns that may emerge in the parent–child relationship and highlights which aspects of these relationships are remembered and narrated by adults. The study is based on a large dataset of 1,592 interview transcripts collected in three languages (English, Dutch, and French), which constitutes an impressive empirical basis for qualitative analysis.

The theoretical background is well developed, and the analytic procedure appears rigorous and carefully conducted. Overall, the study provides a compelling qualitative exploration of schemas that are antithetical to the secure base script.

Importantly, the authors themselves acknowledge that identifying additional schemas or thematic patterns is only the first step. As they note, a crucial next step will be to determine whether these specific schema types can be reliably coded and whether they have unique correlates or predictive value for different developmental or relational outcomes. I agree with this assessment and consider the present paper a meaningful contribution that opens promising directions for future research.

I have only one substantive comment that may help strengthen the manuscript.

The dataset includes transcripts in three different languages (English, Dutch, and French). It would be helpful for readers if the authors provided more information about the composition of the dataset. Specifically, what proportion of the analyzed transcripts came from each linguistic or national sample? And were there any notable differences between these subsamples in terms of the identified schemas or thematic patterns?

While such distinctions may not be central to the qualitative aims of the study, they are nevertheless informative for understanding the scope and diversity of the analyzed material. Additionally, it would be interesting to know whether any attachment-related schemas or themes appeared exclusively or predominantly in one linguistic or cultural context but not in others. Even a brief note addressing this issue would enrich the interpretation of the findings.

Apart from this clarification, I have no major concerns. In my view, the manuscript is already suitable for publication, and addressing the points above would further strengthen an already very interesting and well-executed study.

Reviewer #2: I am truly grateful to the authors for addressing such an important topic. However, I have several major comments that should be addressed before the manuscript can be considered for publication.

Comment 1. The aim of the study could be expanded to include the role of alternative schemas. I suggest that the authors more explicitly highlight the importance of alternative schemas within attachment theory.

Comment 2. The sample is not sufficiently defined or described. I would appreciate it if the authors provided more detailed sample characteristics, including relevant demographic information.

Comment 3. In the discussion section, I recommend elaborating on the theoretical meaning of these alternative schemas in the context of attachment insecurities. Specifically, the implications for different attachment styles should be clarified. How do these findings contribute to our understanding of internal working models?

Comment 4. The manuscript would benefit from a deeper discussion of the theoretical implications of these schemas, particularly in relation to cultural norms.

Comment 5. Finally, I suggest including a clearly defined limitations section that addresses potential issues related to gender, culture, and methodology.

6. PLOS authors have the option to publish the peer review history of their article (what does this mean?). If published, this will include your full peer review and any attached files.

Reviewer #1: No

Reviewer #2: No

---

## [Author Response · Author response to Decision Letter 1]

30 Apr 2026

Response to reviewers

We thank the academic editor and the two anonymous reviewers for their time and guidance following the first review of our paper titled "Organizing Adult Attachment in Alternative Ways: A Qualitative Assessment of Schemas Antithetical to the Secure Base Script." We found their comments helpful and hope they will agree that the paper has been improved. We have addressed each of the reviewers’ comments as follows.

In response to Reviewer 1

1. On the composition of the dataset across linguistic and national samples

‘The dataset includes transcripts in three different languages (English, Dutch, and French). It would be helpful for readers if the authors provided more information about the composition of the dataset. Specifically, what proportion of the analyzed transcripts came from each linguistic or national sample? And were there any notable differences between these subsamples in terms of the identified schemas or thematic patterns?

While such distinctions may not be central to the qualitative aims of the study, they are nevertheless informative for understanding the scope and diversity of the analyzed material. Additionally, it would be interesting to know whether any attachment-related schemas or themes appeared exclusively or predominantly in one linguistic or cultural context but not in others. Even a brief note addressing this issue would enrich the interpretation of the findings.’

We thank the reviewer for the encouraging remarks and thoughtful feedback. We have added the following breakdown of transcripts by language and country of origin, as well as additional demographic information, to the Participants section, page 12 (lines 239 to 253). The section reads:

“As part of the ASPAN network, AAI transcripts were included from English (62%), Dutch (28%), and French-speaking (8%) samples. Approximately half of the Dutch and French transcripts were translated into English by a professional translation service. Transcripts were collected from Canada (8%), the Netherlands (27%), the United Kingdom (5%), and the United States (60%). All interviewees were either parents or expecting a child, with 93% of the total sample female and, on average, 28 years old. A total of 1,592 AAI transcripts collected within ASPAN-CATS were coded for secure base script knowledge by our team of seven trained coders, drawn from both normative (38%) and high-risk populations (62%). High-risk status was defined across contributing studies by the presence of one or more risk factors, including adolescent parenthood, parent psychopathology, parental substance abuse, parental history of abuse or trauma, and demographic risk indicators such as poverty. In terms of ethnicity, 63% of participants identified as Caucasian, 30% as African American, 3% as Hispanic, 1% as Asian, and 2% as other, with 13% reporting immigrant status. Roughly 50% of participants had completed tertiary education.”

Regarding schema differences across linguistic or national subsamples, systematic comparison was beyond the scope of the current study. As a qualitative thematic analysis, this study aimed to identify and characterize the content of alternative schemas in a large, diverse sample, rather than to compare their prevalence or distribution across subgroups. We acknowledge this as a valuable question, however, and note that this is precisely the kind of question the first author intends to pursue in upcoming quantitative work, examining schema prevalence and content across national samples. We have added a brief reference to linguistic diversity in the existing passage in the Reaching Saturation section of the Discussion, page 26 (lines 556-567), which already addresses the broader question of cultural scope and invites future research on this point (additions are underlined). The section reads:

“We reached a saturation of themes for alternative schemas during coding of this large corpus of AAIs, further increasing confidence in the schemas identified. After a period with no additional themes emerging beyond Favoritism, Restrictive, and Incompetent, the coding team conducted a saturation check at 814 transcripts coded. No evidence for additional novel themes emerged beyond this point. Although this may indicate that the sample was sufficiently diverse to cover the variation present in Western, English-dominant samples, novel themes may occur in other linguistic contexts or cultural populations. The attachment field has been widely criticized for its assumptions of universality in attachment (i.e., Behrens, 2016; Thompson, 2019). Indeed, it is possible that certain elements assumed to be central to secure base provision in Western society may not necessarily be present in other cultures.”

In response to Reviewer 2

1. On expanding on the role of alternative schemas and their importance within attachment theory in the Study Aims

‘The aim of the study could be expanded to include the role of alternative schemas. I suggest that the authors more explicitly highlight the importance of alternative schemas within attachment theory.’

In response to Reviewer 2's feedback, we have expanded on the theoretical significance of alternative schemas and their place within the broader attachment framework. We have added the following paragraph to the theoretical section on Scripts and Schemas on pages 8 and 9 (lines 185-199) that further underscores the potential impact of differentiating among the alternative schemas:

“Empirical work examining the antecedents and outcomes of alternative schemas has yet to follow. Measured using the ASA, secure base script knowledge has been linked to the quality of caregiving received in childhood and to later adult depression symptoms and romantic relationship quality (e.g., Dagan et al., 2021; Steele et al., 2014), underscoring how early caregiving experiences may become encoded in representational scripts that carry consequences across the lifespan. Emerging work examining the impact of maternal childhood abuse versus neglect on infant brain development further suggests that different types of adverse caregiving experiences may have distinct developmental consequences (Lyons-Ruth et al., 2023). This raises the question of whether such experiences also give rise to qualitatively different representational schemas in the adult mind, as captured in the AAI. Whether the perceived relational dynamics captured by alternative schemas, rather than broader classifications, carry differential relevance for such outcomes remains an open and important question. Before such empirical questions can be addressed, a necessary first step is to systematically characterize and refine the set of alternative schemas amongst large, risk-diverse samples.”

We have also added some text to the ‘Current Study’ paragraph on page 10 (lines 205-212) to clarify the role of alternative schemas and strengthen our overall aim (additions are underlined):

“In trying to understand how individuals make meaning of childhood experiences with intrusive or distressing caregivers, it is important to consider that the absence of secure base expectations may not simply indicate a lack of representational structure, but may also reflect the presence of alternative ways of organizing expectations about attachment relationships. The purpose of the current study was to explore the content and scope of alternative schemas in the AAI, as a step towards understanding how characterizations of unpredictable and distressing caregiving in childhood may become internalized as cognitive blueprints in adulthood. We therefore conducted a systematic, semi-open assessment of the preliminary set of alternative schemas within a large, risk-diverse set of 14 samples that are part of the Attachment Secondary Processing and Analysis Network (ASPAN). We first evaluated whether the set of alternative schemas identified by Waters and Facompré (2021) also occurred in the current dataset in the form originally described. Second, we examined whether novel themes could be identified, justifying the addition of novel alternative schemas to the set. This broader set of alternative schemas might contribute to further nuancing our understanding of attachment representations, particularly in individuals who experienced adverse caregiving environments.”

2. On providing further details on sample characteristics

‘The sample is not sufficiently defined or described. I would appreciate it if the authors provided more detailed sample characteristics, including relevant demographic information.’

We have included additional sample characteristics and demographic characteristics on page 12 (lines 239-253). Please see our Response to Reviewer 1, comment 1.

3. On the implications of alternative schemas for attachment insecurities and attachment styles

“In the discussion section, I recommend elaborating on the theoretical meaning of these alternative schemas in the context of attachment insecurities. Specifically, the implications for different attachment styles should be clarified. How do these findings contribute to our understanding of internal working models?”

We have added a new subsection titled Theoretical meaning of alternative schemas to the Discussion on pages 28-29 (lines 605-631), in which we elaborate on the theoretical implications of alternative schemas in the context of non-autonomous attachment representations. We note that in line with AAI convention, we use the term ‘non-autonomous’ rather than ‘insecure’ throughout the manuscript, or specify the dismissing or preoccupied representations.

In this new subsection, we position alternative schemas as offering a complementary lens to existing AAI classifications: rather than assessing non-autonomous attachment through discourse incoherence or attentional strategies, alternative schemas capture the explicit content of attachment-relevant experiences as recalled during the AAI. We further elaborate on how mapping alternative schemas onto non-autonomous classifications may help specify which relational expectations and experiences underlie dismissing or preoccupied attachment strategies. The section reads:

“Theoretical meaning of alternative schemas

Beyond reliability and coding utility, capturing distinct schemas may carry theoretical meaning in its own right. These schemas reflect a qualitatively distinct representational organization of attachment-relevant experiences with primary caregivers. Rather than simply indicating the absence of secure base script knowledge, they outline in detail the recurring expectations of how caregiving relationships function under conditions of fear, distress, role confusion, unpredictability, or emotional unavailability.

This distinction may offer a complementary approach to understanding non-autonomous attachment classifications. With the move to the level of representation (Main, Kaplan, & Cassidy, 1985), security and insecurity in adult representations of attachment are assessed on the basis of discourse coherence and attentional strategies during the AAI, where dismissing and preoccupied classifications describe ways in which these discourses of attachment-related experiences are limited and distorted. Alternative schemas offer a different lens: rather than how narratives are organized, they capture the explicit content of attachment-relevant experiences as recalled during the AAI. Mapping the alternative schemas onto the non-autonomous classifications may help capture which expectations and behaviors underlie preoccupied or dismissing attachment strategies. For instance, dismissing representations may not only manifest through implicit discourse indicators such as idealization or lack of memory (Hesse, 2008), but also through explicit, temporal-causal, relational content described by schemas such as Self-Involved/Neglect or Incompetent. Similarly, discourse indicators for preoccupied (e.g., passive speech, difficulty moving on from topics) may be further specified through schemas such as Enmeshed or Role Reversal. Examining this additional layer of specificity, detailing the particular parenting behaviors that led to the caregiver being perceived as a source of distress rather than support, may hold relevance for both research and clinical practice. Empirical work directly examining the overlap between alternative schemas and AAI classifications is needed to test these propositions.”

Furthermore, we use the term "representation" rather than "internal working model" throughout the manuscript, as this aligns more closely with the theoretical framework of the secure base script, within which alternative schemas are embedded. Similarly, we situate our findings within the AAI-based classification tradition of attachment representations rather than the self-report attachment styles literature. However, we acknowledge that the question of how alternative schemas relate to romantic relationship quality, among other spheres of adult life, is an important direction for future work. We have expanded the Antecedents, Consequences, and Utility section on pages 29 and 30 (lines 643-675) accordingly to address these implications:

“The relevance of such questions is underscored by existing work on secure base script knowledge more broadly. Dagan and colleagues (2021, 2024, 2025) showed that greater secure base script knowledge was associated with better emotional and physical health, as well as higher romantic relationship quality, specifically lower perceived hostility from a romantic partner. Furthermore, secure base script knowledge mediated the relation between maternal and paternal sensitivity in childhood and romantic partner hostility and emotional health. This finding on secure base script knowledge aligns with previous research, supporting the idea that this form of attachment representation, built on repeated experiences of sensitive caregiving (Steele et al., 2014; Nivison et al., 2020; Schoenmaker et al., 2015; Waters et al., 2017), may influence the processing of interpersonal information and behaviours later in life (Collins & Sroufe, 1999; Dykas & Cassidy, 2011; Dagan et al., 2024; Waters et al., 2013, 2018). Considering that alternative schemas similarly reflect expectations based on childhood experiences with primary caregivers, it remains an open question how these schemas, which specify content that conflicts with elements of secure base provision, may also impact adult functioning.

The alternative schemas may offer both research and practical utility by providing detailed information on how adults make sense of caregiving relationships and organize their broader expectations of others. What it means, representationally, to organize expectations around experiences of threat and fear may be vastly different from experiences of taking on the caregiving role: one might expect that an individual describing the Harsh and Threatening Parenting schema perceives the world as a particularly hostile place, in contrast to someone with a schema scored as Enmeshed. Gaining a better understanding of the antecedents and consequences of such cognitive schemas may therefore be a promising avenue for both researchers and clinicians. In this regard, Raby and colleagues (2021) showed that secure base script knowledge improved following intervention in parents referred by Child Protective Services, and that this increase mediated improvements in parental sensitivity seven years later. These findings support the view that cognitive scripts are malleable and can be modified through new experiences (Bowlby, 1988; Schank & Abelson, 1977; Waters et al., 2019; but see Witte et al., 2024, for null effects on script change following intervention, which the authors attribute to possible sleeper effects). The current study offers further description and delineation of alternative schemas in a risk-diverse sample, providing a first step towards future research on whether cognitive scripts centered on themes of fear and unpredictability may also show malleability and openness to change.”

4. On theoretical implications of alternative schemas in relation to cultural norms

‘The manuscript would benefit from a deeper discussion of the theoretical implications of these schemas, particularly in

---

## [Editor Report · Decision Letter 1]

4 May 2026

Organizing Adult Attachment in Alternative Ways:

A Qualitative Assessment of Schemas Antithetical to the Secure Base Script

PONE-D-26-03584R1

Dear Dr. Maingot,

We’re pleased to inform you that your manuscript has been judged scientifically suitable for publication and will be formally accepted for publication once it meets all outstanding technical requirements.

Kind regards,

Tobias Otterbring

Academic Editor

PLOS One

Additional Editor Comments (optional):

Dear authors,

I thank you for delivering a carefully crafted revision. As I believe you have compellingly addressed all the substantive concerns raised by the reviewers, I am happy to recommend acceptance of your work in its current form. Congratulations!

Kind regards,

Tobias Otterbring

Handling Editor, PLOS One
---

## [Editor Report · Acceptance letter]

PONE-D-26-03584R1

PLOS One

Dear Dr. Maingot,

I'm pleased to inform you that your manuscript has been deemed suitable for publication in PLOS One. Congratulations! Your manuscript is now being handed over to our production team.

Kind regards,

on behalf of

Professor Tobias Otterbring

Academic Editor

PLOS One